# Influence of the use of an adhesive connection on the joint strength of modular hip endoprostheses

**Ann-Kathrin Einfeldt[1], Beate Legutko[2], Philipp-Cornelius Pott[3], Benjamin Bergmann[2], Berend Denkena[2], Christof Hurschler[1], Bastian Welke[1]***

1 Laboratory for Biomechanics and Biomaterials, Department of Orthopedic Surgery – Annastift DIAKOVERE, Hannover Medical School, Hannover, Germany, 2 Institute of Production Engineering and Machine Tools, Leibniz University Hannover, Garbsen, Germany, 3 Department of Prosthetic Dentistry and Biomedical Materials Research, Hannover Medical School, Hannover, Germany

* Welke.bastian@mh-hannover.de

## Abstract

### Introduction

Modular hip implants enables a more precise adaptation of the prosthesis to the patient's anatomy. However, they also carry the risk of increased revision rates due to micromotion at the taper junction. In order to minimize this risk, one potential solution is to establish an adhesive bond between the metal taper junctions. Load-stable bonding techniques, already successfully employed in dentistry for connecting materials such as metals and ceramics or different alloys, offer a promising approach. Nevertheless, the bond strength of tapered adhesive bonds in modular hip implants has not been investigated to date.

### Materials and methods

Twenty-eight tapered junctions, consisting of a taper (female taper) and a trunnion (male taper) were turned using TiAl6V4 ELI (n = 16) and CoCr28Mo6 (n = 12). The process parameters cutting speed ($v_c$ = 50 m/min or 100 m/min) and feed (f = 0.1 mm, 0.05 mm or 0.2 mm) were varied for the trunnions. For each set of process parameters, one trunnion and one taper were additionally subjected to sandblasting. To investigate the effect of geometry, angular mismatch in the samples were measured. The taper pairs were bonded with a biocompatible adhesive, and push-out tests were subsequently performed.

### Results

The push-out forces generated from the taper connections where both tapers were sandblasted showed a mean push-out force of 5.70 kN. For the samples with only the trunnion sandblasted, the mean force was 0.58 kN, while for the samples with only taper sandblasted the mean push-out force was 1.32 kN. When neither of the tapers was sandblasted the mean push-out force was 0.91 kN. No significant effect of the process parameters on the push-out force was observed. Only the reduced valley depth Svk showed a slight correlation for the CoCr28Mo6 samples ($R^2$ = 0.54). The taper pairs with taper mismatch (between

**Data Availability Statement:** All relevant data are within the manuscript and its Supporting information files.

**Funding:** The study was financially supported by the Deutsche Forschungsgemeinschaft (DFG, German Research Foundation) within the SFB/TRR-298-SIIRI-Project-ID 426335750 "Safety integrated and infection reactive implants" in subproject A04. Publication costs are covered by the German Research Foundation (DFG) and the Open Access Publication Fund of Hannover Medical School (MHH). The funders had no role in study design, data collection and analysis, decision to publish, or preparation of the manuscript.

**Competing interests:** The authors have declared that no competing interests exist.

**Abbreviations:** CCD, caput-collum-diaphyseal; f, feed; Rv, maximum depth of profice; Rz, maximum height of profile; Sb, sandblasted; Spk, reduced peak hight; Svk, reduced valley depth; THA, Total hip arthroplasty; $v_c$, cutting speed.

trunnion and taper) greater than |0.1˚| did not show lower push-out forces than the specimens with lower taper mismatch.

## Conclusions

Sandblasted and adhesive-bonded tapered connections represent a viable suitable alternative for modular hip implant connections. Slight differences in taper geometry do not result in reduced push-out forces and are compensated by the adhesive. In mechanically joined tapers these differences can lead to higher wear rates. Further investigation under realistic test conditions is necessary to assess long-term suitability.

## Introduction

Total hip arthroplasty (THA) is one of the most common inpatient surgeries, with approximately 240,000 procedures performed each year [1]. Since the late 1990s, modular hip endoprostheses have allowed surgeons to better address individual patient characteristics by using specific implant components with different stem neck sizes and angles [1, 2]. For example, bi-modular tapered neck adapters allowed the generation of different antetorsion and caput-collum-diaphyseal (CCD) angles, facilitating optimal reconstruction of the biomechanics of the joint [1].

Nonetheless, the advantages of bi-modularity were associated with increased complications, resulting in revision rates of 6% compared to 3% for monolithic prosthesis systems after 5 years [3]. Typical damage observed is the clinical failure of tapered stem junctions due to micromotion, which appear to be dependent on the design and material coupling (Co-Cr29-Mo vs. Ti-6Al-4V) at the junction between the stem and the adapter [2, 4]. For instance, titanium neck adapters exhibited significantly larger micromotions than cobalt-chromium adapters and contaminated interfaces also showed significantly larger micromotions [4]. These micromotions can lead to fretting and crevice corrosion at the taper junction, potentially causing premature connection failure accounting for 2% to 3% of all THA revisions [1, 5].

Despite being a well-known phenomenon the exact causes of theses complications remain poorly understood. However, a number of mechanisms and possible causes are commonly cited, such as galvanic and crevice corrosion, micromotions and fretting, and contamination of the joint leading to the release of metal ions [6]. Furthermore, modularity plays a central role in the damage tolerance of implants, and each additional modular connection therefore increases the effect.

Another aspect leading to higher wear rates during dynamic loading is a so-called taper mismatch [7, 8]. It describes the difference of the angle between the trunnion and the taper of the hip implant and is sometimes designed into the systems to ensure consistent performance [8]. Nevertheless, a few finite element studies found out that smaller mismatches result into less wear in modular hip implants [7, 8]. Taper mismatch can be negative (larger taper than trunnion), or positive (smaller taper than trunnion). Ashkanfar et al. performed a comparative FEM study of the effect of different taper mismatch angels and suggested that taper mismatch should be manufactured with to less than 6' or 0,1˚ [9].

The above mentioned possible causes of taper junction failure area multiple and have diverse causes. Despite their utility in restoring patient specific geometries, bi-modular hip implants have widely been recalled from the market due to the relatively high rates of failure

observed. One possible method to address the aforementioned failure mechanisms (micromotion, fretting-corrosion, contamination, and taper-missmatch tolerances) may be to establish an adhesive bond in the metal taper junctions. Such an adhesive bond has the potential to enhance the mechanical load transfer between the metal taper partners and to prevent corrosion and release of particles and ions into the joint. These types of adhesively joined interfaces are not new. In dentistry, adhesive techniques have been well established for bonding various materials in different types of dentures and crowns [10–13]. Thus, titanium and cobalt-chromium tapered bonds have successfully been utilized and investigated in this context [14, 15]. Furthermore, several bonding systems have been reported and the use of adhesives proven to be effective for bonding titanium alloys [15]. Bonding systems investigated in this context have included different primers and luting agents, often in conjunction with surface modification such as airborne-particle abrasion, a specialized surface treatment technique similar to sandblasting [16]. The dental application most comparable to the taper-junctions addressed in this study are hybrid abutments, where the respective restoration is bonded to standardized implant abutments [17]. However, the bond strength of tapered adhesive bonds as used in in modular hip implants has not been investigated to date. Due to the high revision rates in bimodular hip implants, they widely disappeared from the market. Nevertheless, this study aims to investigate adhesive bonding with the aim of addressing the issues that led to their failure. The aim of this study was therefore to perform an exploratory investigate of the possible suitability of using an adhesive bond to join modular hip implant junctions. To achieve this goal, various process parameters and the resulting surface topographies of titanium and cobalt chromium bonded joints were examined in relation to their bond strength. We hypothesize that a bonded connection with a rough surface achieves at least the same strength as conventionally joined implant junctions.

## Materials and methods

### Specimen preparation

In total 28 taper connections consisting of a trunnion (male taper) and a taper (female taper) were manufactured. The tapers were manufactured on a linear lathe (CTX 420, Gildemeister, Wernau, Germany) and the trunnions on the turnmill center (NTX 1000, DMG MORI, Wernau, Germany). For turning the trunnion, indexable inserts (type VBMT 1604 MM 1115, Sandvik Tooling Deutschland GmbH, Duesseldorf, Germany) were used. The trunnion has a nominal taper angle of $\theta_{AK} = 11.33°$, as well as the diameters of $d_E = 12.74$ mm and $d_H = 14.88$ mm. The length of the angular surface in relation to the central axis of the trunnion was $l_{AK} = 11.22$ mm (Fig 1A) and the length of the bore of the taper was $l_{IK} = 14$ mm (Fig 1B) to ensure that in case of minor deviations of the geometry due to the manufacturing process, the entire taper length is still in contact at the junction.

The process parameters cutting speed ($v_c = 50$ m/min or 100 m/min) and feed (f = 0.1 mm, 0.05 mm or 0.2 mm) were varied for the trunnions, resulting in a total of seven different testing groups, each consisting out of four taper pairs (Table 1). In the first four groups the taper pairs (Ti 1 –Ti 4) were made out of TiAl6V4 ELI whereas in the remaining three groups the samples were manufactured out of CoCr28Mo6 (CoCr 5-CoCr 7). The depth of cut $a_p = 0.25$ mm were kept constant. The process parameters for the tapers were not varied to ensure that all observed effects were dependent on the surface topography of the trunnions. The tapers were manufactured with indexable inserts (type CCMT 060204 Sandvik Tooling Deutschland GmbH, Duesseldorf, Germany) on a boring bar. The machining process was performed with $v_c = 70$ m/min, f = 0.1 mm and ap = 0.25 mm. In each testing group one trunnion and one taper were additionally sandblasted (sb). In groups Ti 2, Ti 4, CoCr 5 and CoCr 7 both tapers of a taper

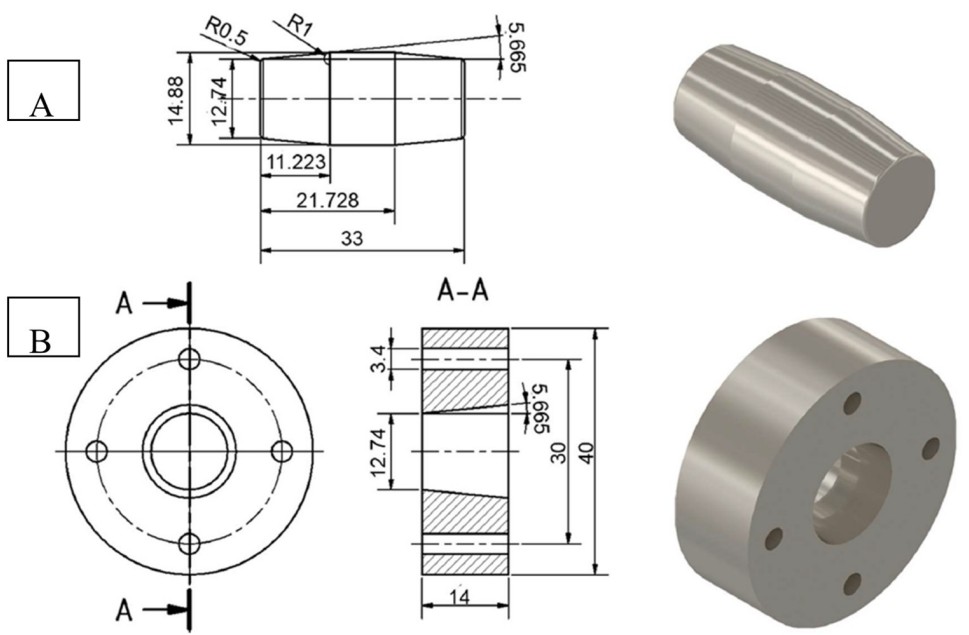

**Fig 1. Technical drawing of the trunnion (A) and the taper (B).**

pair were sandblasted whereas in the remaining three groups different tapers were sandblasted (S1 Table). The sandblasting pressure was set to p = 2.3 bar, and the taper contact surface was manually blasted with a grain size of x = 101 μm corundum at approximately 10 mm distance from the surface for 30 seconds.

## Measurement of the samples

As possible angular mismatch of the tapers can also influence the separating forces that occur in the taper connections, a 3D coordinate measurement system (Reference Xi 1076, Leitz, Stockholm, Sweden) was used to determine the resulting geometry of the manufactured tapers.

Subsequently, roughness measurements were performed for the trunnion and tapers. The tapers were scanned over a distance of 8 mm (MarSurf LD 130 with the LP C 3.5-102/90˚ probe, Mahr, Göttingen, Germany). The probe then repeated this process a further nine times at different starting points in order to obtain a more accurate representation of the surface topography across a wider measuring distance.

**Table 1. Overview over the different process parameters for each taper group.** Tapers in group 1–4 were manufactured from TiAl6V4 ELI, tapers in group 5–7 were manufactured from CoCr28Mo6 (shaded in grey).

|  | $v_c$ [m/min] | f [mm] |
|---|---|---|
| Ti 1 | 50 | 0.10 |
| Ti 2 | 100 | 0.05 |
| Ti 3 | 50 | 0.05 |
| Ti 4 | 100 | 0.20 |
| CoCr 5 | 50 | 0.10 |
| CoCr 6 | 100 | 0.05 |
| CoCr 7 | 50 | 0.05 |

The roughness measurement of the trunnions was conducted using an optical measuring device over a length of 5 mm (Duo Vario, Confovis, Neu-Isenburg, Germany). Measurements were taken for all tapers, starting from the main diameter in the direction of the end diameter and from the end diameter in the direction of the main diameter. This approach resulted in a total measuring length of 10 mm per taper, allowing for the analysis of potential deviations in the surface over the entire contact area of the taper.

The measurement data was analysed using the MountainsMap analysis programme (Digital Surf, 8.2, Besançon, France). The characteristic values of the surface topographies were determined. The roughness parameters maximum height of profile (Rz) and maximum depth of profile (Rv) according to DIN EN ISO 21920 and the primary surface topography profile (Reduced Peak Hight (Spk), Reduced Valley Depth (Svk)) according to DIN EN ISO 25178, were recorded over a length ranging from 3.8 to 4.6 mm.

## Bonding the tapers

After manufacturing and blasting, the tapers were bond with a biocompatible adhesive system. The adhesive is a self-curing luting composite used for the adhesive luting of metal and metal-ceramics, among other materials. The adhesive system used (Multilink Automix, Ivoclar Vivadent AG, Ellwangen, Germany) is a self-curing material consisting of a 2-component primer system (Multilink Primer A + B) and a 2-component luting resin. Before bonding, the tapers were cleaned in an ultrasonic bath with alcohol to eliminate any negative influences of possible contamination on the bond.

The two primer components A and B were mixed in a 1:1 ratio and evenly applied to the contact surfaces without the formation of drops until the surface was completely wetted. The exposure time for the primer was 2 minutes. The surfaces that needed to be free of the luting resin were isolated with cling film. The luting resin was then applied bubble-free into the opening of the taper using the application cannula and the tapers were joined together (Fig 2). The

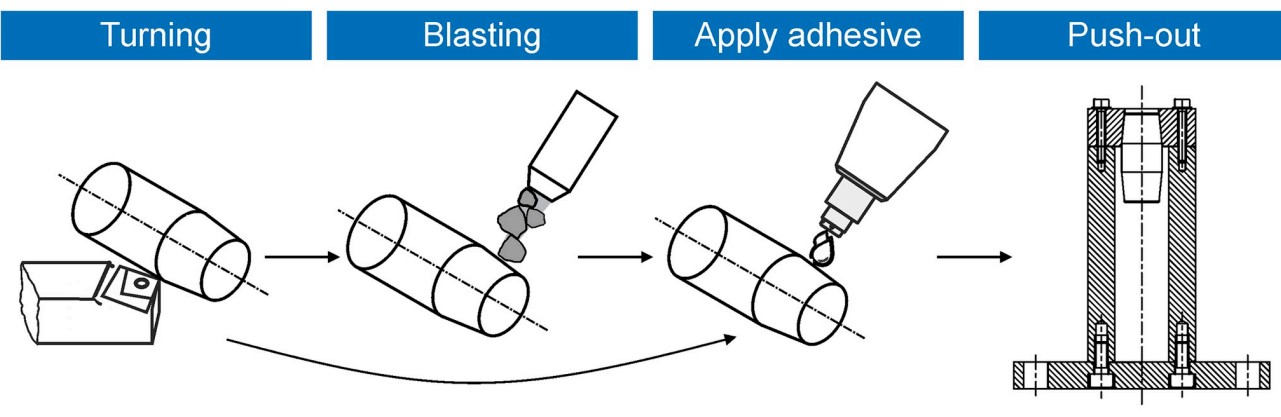

**Turning process:** $v_C$ = var. f = var. $a_p$ = 0.25 mm

**Material:** TiAl6V4 ELI, CoCrMo28 **Blasting material:** Corundum, d = 101 µm, p = 2.3 bar

**Primer:** Multilink A+B **Adhesive:** Multilink Automix Ivoclar

**Separating:** Speed: 8 µm/s **Force attack:** centric

Leg/113512 © IFW

**Fig 2. Schematic illustration of the individual processing steps and the setup for the push-out tests.**

trunnion was inserted into the taper and allowed to sag under its own weight until the adhesive oozes out at the edge of the contact surface. This excess adhesive can be removed within the curing time of 8 to 10 minutes at 23°C ± 1°C (Fig 2).

## Push-out tests

The push-out tests were performed using a universal testing machine (8501, Instron, Illinois Tool Works Inc., Norwood, USA). The tests were carried out after a curing time of 24 hours at 23°C ± 1°C of the probes. The machine's stamp moved down hydraulically at a velocity of 0.48 mm/min, pressing the trunnion out of the taper. At every 0.001 mm displacement of the hydraulic cylinder, the following parameters were recorded: time, axial force and axial displacement. The measured displacement of the testing machine was subsequently corrected for the rigidity of the setup.

## Results

### Taper geometry

The main diameter of the trunnion was targeted to be $d_M$ = 14.88 mm, the end diameter $d_E$ = 12.74 mm, the taper length l = 11.22 mm, and the angle $\theta_n$ = 11.33°.

All trunnions show deviations from the nominal manufacturing values, with a mean deviation of all trunnions was 0.18 ± 0.19 mm, whereas the Ti 3-row showed the smallest deviations (Deviation $d_M$ = 0.06 mm). The TiAl6V4 ELI tapers tend to be above, and the CoCr28Mo6 tapers below the nominal values. The mean deviation of the end diameter for all trunnions was 0.22 ± 0.23 mm, and for the taper length 0.03 ± 0.03 mm. The Ti 2-row shows the smallest deviations from the nominal value for both parameters (Deviation $d_E$ = 0.04 mm, Deviation l = 0.004 mm). CoCr28Mo6 tapers showed greater dispersions than the TiAl6V4 ELI tapers. The mean angle deviation of all trunnions is 0.04 ± 0.02°, with the Ti 3-row again showing the smallest deviations (θ = 0.02°) from the nominal manufacturing value whereby and the CoCr28Mo6 tapers exhibiting greater deviations than the TiAl6V4 ELI tapers.

The face diameter of the taper was aimed to be $d_F$ = 14.88 mm, the end diameter $d_E$ = 12.74 mm and the angle $\theta_n$ = 11.33°.

All face diameters of the tapers are above the nominal value (Deviation $d_F$ = 0.11 ± 0.02 mm) whereas in the end diameter they are below the nominal value (Deviation $d_E$ = 0.37 ± 0.02 mm). The mean deviation of the nominal angle of all tapers is 0.64 ± 0.02° and is above the nominal value for all tapers.

Due to the deviations from the nominal values different free volumes—areas where there is no contact between the tapers—exist for the taper pairs. All 28 taper pairs exhibit a negative taper mismatch (trunnion angle < taper angle) and therefore a proximal contact. All taper pairs show a taper mismatch greater than |0.1°| with a mean of -0.182 ± 0.047° (range -0.012° to -0.369°) (S1 Table).

### Roughness

The roughness measurement of the trunnions after sandblasting shows the greatest roughness in the Ti 4-group, followed by the CoCr 5 and Ti 1 groups (Table 2). CoCr28Mo6 tapers exhibit greater Rz and Rv values than TiAl6V4 ELI tapers when the same process parameters are used. Ti 4 group reaches the greatest Spk, followed by the Ti 1 and the CoCr 5 group. CoCr 5 group show the greatest Svk.

After sandblasting, the trunnions show a more porous surface. For the TiAl6V4 ELI tapers the Rz and Rv values hardly change whereas for the CoCr28Mo6 tapers these values were

**Table 2. Overview over the different roughness and process parameters for each sample group of trunnions before (shaded in blue) and the one trunnion per sample group after sandblasting.**

| | Mean Rz [μm] | Mean Rv [μm] | Mean Spk [μm] | Mean Svk [μm] | Rz [μm] | Rv [μm] | Spk [μm] | Svk [μm] |
|---|---|---|---|---|---|---|---|---|
| Ti 1 | 4.812 | 2.335 | 1.218 | 0.685 | 4.531 | 2.014 | 1.437 | 0.932 |
| Ti 2 | 1.892 | 0.968 | 0.330 | 0.396 | 2.027 | 1.118 | 1.054 | 0.983 |
| Ti 3 | 2.412 | 1.221 | 0.615 | 0.394 | 1.495 | 0.817 | 1.004 | 1.085 |
| Ti 4 | 15.080 | 5.470 | 8.554 | 2.326 | 12.500 | 4.705 | 6.625 | 1.115 |
| CoCr 5 | 5.057 | 2.673 | 0.837 | 1.316 | 6.595 | 4.164 | 0.798 | 3.035 |
| CoCr 6 | 2.606 | 1.274 | 0.475 | 0.375 | 2.175 | 1.090 | 1.010 | 0.744 |
| CoCr 7 | 2.854 | 1.508 | 0.480 | 0.689 | 2.488 | 1.515 | 0.813 | 1.331 |

Rz = maximum height of profile; Rv = maximum depth of profile; Spk = Reduced Peak Hight; Svk = Reduced Valley Depth.

reduced after sandblasting. Both the Spk and the Svk increases after sandblasting for both materials.

As there was no variation in the process parameters for the tapers, only differences between the sandblasted and not sandblasted tapers were evaluated. The sandblasted surface is more porous (Fig 3) leading to increased roughness parameters compared to the untreated surface.

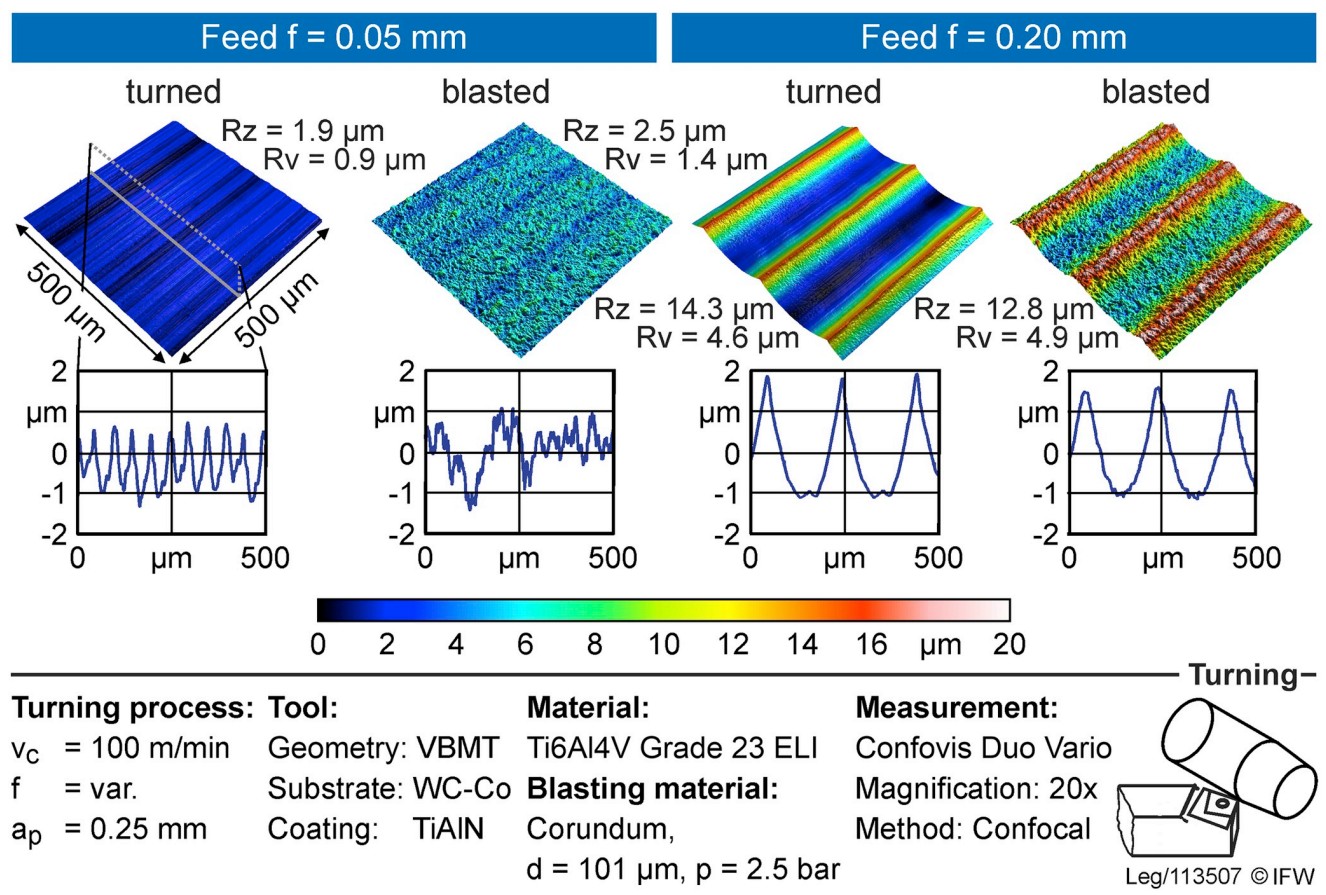

**Fig 3. Surface and profile plots of a turned sample (left in each case) and a turned and subsequently sandblasted sample (right in each case) of a trunnion.**

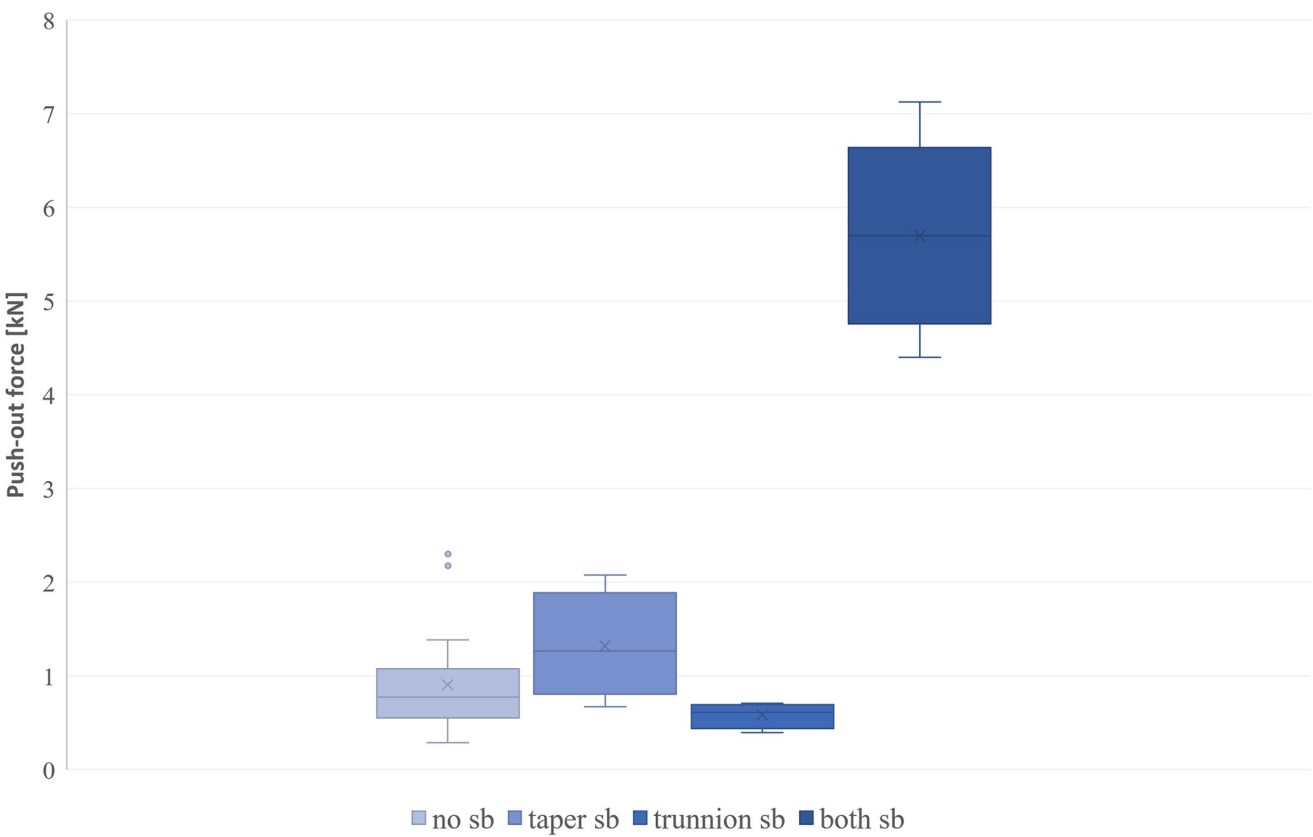

**Fig 4. Results of the push-out test for all taper connections and materials divided into groups by the sandblasting (sb) situation.** Values above and below 1.5 times the interquartile range are shown as outliers.

## Push-out tests

The results of the push-out tests show the greatest forces (mean$_F$ = 5.70 kN) for the taper connections where both tapers were sandblasted (Fig 4). All other pairing situations show smaller forces (only taper sandblasted mean$_F$ = 1.32 kN, no sandblasting mean$_F$ = 0.91 kN), with the smallest force when only the trunnion was sandblasted (meanF = 0.58 kN).

To investigate whether the mismatches in taper geometry influence the resulting push-out forces the free volume for each taper pair was calculated. The calculated free volume includes the taper mismatch and the roughness profile resulting from the machining process of the trunnion and the taper. No correlations between free volume and therefore the adhesive volume and the push-out forces could be seen.

When looking at the push-out forces in relation to the different process parameters and therefore the surface topographies, no correlation between the roughness parameters Rz and Rv and the push-out force were observed ($R^2_{Rz}$ = 0.03; $R^2_{Rv}$ = 0.08).

No correlations were observed ($R^2_{allsamples}$ = 0.02, $R^2_{Ti}$ = 0.02, $R^2_{CoCr}$ = 0.13) for the push-out forces in relation to the tip height. For the Svk values, a slight correlation ($R^2_{CoCr}$ = 0.54, p = 0.08) can be observed especially for the CoCr28Mo6 samples (Fig 5).

The force-distance curves of the push-out tests are exemplarily shown for individual samples of each group (see Fig 6). The graph shows the smallest incline for the only turned CoCr28Mo6 sample (Fig 6, red line). For the only turned TiAl6V4 ELI sample the incline is

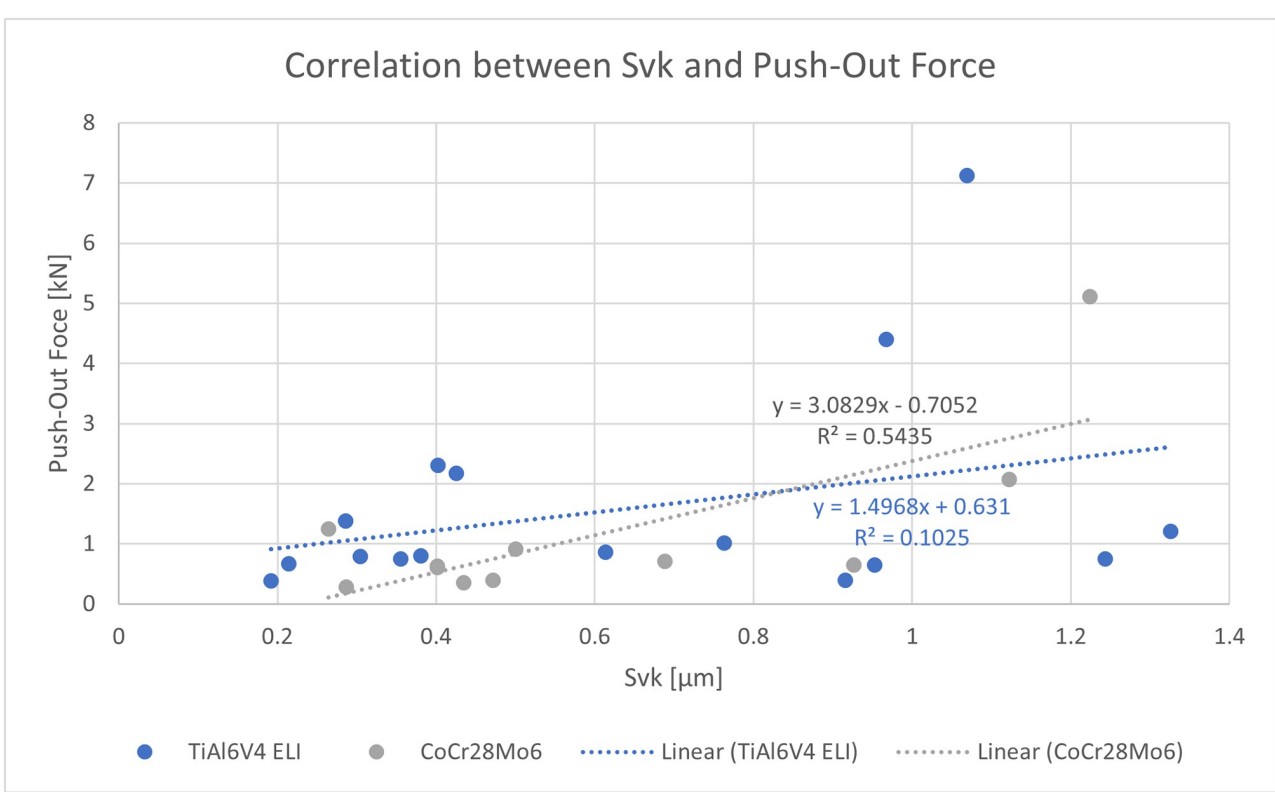

**Fig 5. Correlation between Svk (Reduced Valley Depth) and push-out force for all TiAl6V4 ELI samples (blue) and CoCr28Mo6 samples (grey).** Dotted line shows the resulting regression line for each material type.

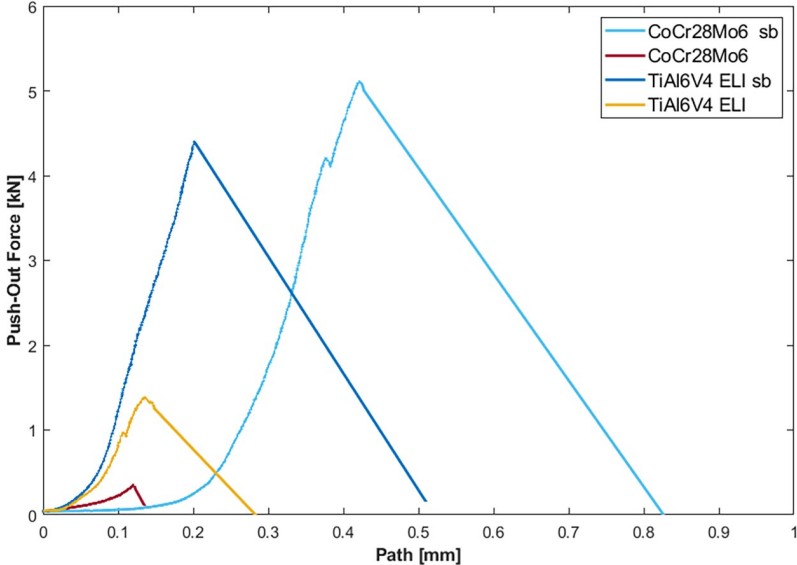

**Fig 6. Force-path curves of the push-out tests exemplarily shown for individual samples of the different groups (CoCr28Mo6 only turned = red; TiAl6V4 ELI only turned = yellow; CoCr28Mo6 both sandblasted = light blue; TiAl6V4 ELI both sandblasted = dark blue.** The measured displacement of the testing machine was corrected for the stiffness of the setup.

steeper ([Fig 6](), yellow line). When both tapers of one sample are sandblasted the steepest inclination profiles are seen, ([Fig 6](), light and dark blue). When one sample is sandblasted and one sample is turned, no clear inclination profile can be observed for the samples.

A visual analysis of the tapers after the push-out tests showed, that the adhesive adheres only to the sandblasted surfaces. No adhesive residues were observed macroscopically at the surfaces that were only turned.

## Discussion

### Taper geometry

The smallest deviations in taper geometry could be observed when the cutting speed and the feed were both small as shown in the Ti_3 group with a $d_M$ = 0.06 mm ($v_c$ = 50 m/min and f = 0.05 mm). The feed seems to have a greater influence on the precision of the taper geometry than the speed, as both groups with a feed of 0.05 mm (Ti_2 and Ti_3) show the smallest deviations from nominal values even though different speeds were used. CoCr28Mo6 trunnions tend to be below the nominal values whereas TiAl6V4 tend to be above them. The tapers show that the face diameter of all tapers lies above whereas the end diameter lies below the nominal value. This is due to the different tool deflection caused by the different material properties.

When comparing the taper material, CoCr28Mo6 shows greater deviations from the nominal values than TiAl6V4 ELI. As CoCr28Mo6 is harder than TiAl6V4 ELI, this could be the reason. Overall, the results show that the material has a greater influence on the deviations on taper geometry than the process parameters.

All 28 samples show a proximal contact area. Bitter et al. [8] recommend that a distal contact area should be chosen to reduce the wear rate. As there only samples showing a proximal contact area no conclusion can be drawn about which contact area is to be preferred in adhesive bonded taper pairs.

All taper pairs show a taper mismatch greater than |0.1˚|. The results of the push-out tests indicate that there is no correlation between the amount of mismatch and the push-out force. Therefore, the adhesive appears to compensate slight differences in taper geometry and consequently taper mismatch.

### Roughness

Most stem tapers that are turned have trough to peak heights form 10 to 13 μm. In order to investigate the influence of the surface topographie on the push-out force the range of the surface roughness is from Rz = 5 μm to Rz = 15 μm. CoCr28Mo6 was observed to have greater Rz and Rv roughness than TiAl6V6 ELI. Reduced peak height on the other hand was greater for Ti than for CoCr28Mo6. Before sandblasting, it could be shown that the feed has the greatest influence on the roughness, whereas the speed does not seem to influence this parameter.

After sandblasting, the roughness parameters hardly change for the TiAl6V6 ELI trunnions whereas for the CoCr28Mo6 tapers roughness is reduced. It cannot be ruled out that this difference is due to the manual sandblasting, which might have led to inconsistent influences on the surface. When looking at the surface topographies before and after sandblasting it is clear that the surface topography after sandblasting is more stochastic and the possible bonding surface is enlarged. For tapers the roughness increases after sandblasting regardless of the material.

## Push-out tests

A preliminary study on taper junctions of the same geometry and materials showed that tapers with an assembled at a speed of v = 8 μm/s and a force of up to F = 4 kN reached push-out forces between Fseparate = 0.79 and 1.8 kN [18]. The push-out tests in this study show that taper connections where both tapers were sandblasted before bonding have the greatest push-out forces, ranging from 4.4 to 7.1 kN. These adhesively bonded junctions thus exceeded native taper junctions by a minimum of 244%. Other studies of native taper pull out force have reported ranges of pull-off force under normal conditions for different material pairings and surface modifications ranging from 1.6 kN to 3 kN [19–21], a study using very high impaction forces of 14kN reported pull-off forces of up to 6 kN [22]. Mechanically, the push-out test we performed is equivalent to the pull off test as described in the cited studies.

The Svk shows a slight correlation with the resulting push-out force especially for the CoCr28Mo6 tapers whereas all other roughness parameters are not showing any correlations. The greater the Svk the more liquid and in this case, adhesive, can be collected on the surface of a sample. A greater amount of adhesive on the surface therefore leads to increased bonding strength. After sandblasting this parameter increases for all the samples. This is in accordance with the increased push-out forces for the sandblasted taper connections. Also, the force-displacement curves and the visual analysis of the tapers support this thesis. The steeper inclination of the force time curve for the sandblasted samples, indicate a stiffer junction of the taper pairs as the adhesive adheres to both taper partners. Sandblasting leads to a more porous surface where the adhesive is more likely to adhere to than to only turned samples. This could be seen in the visual analysis after the push-out test.

Even though the other roughness parameters did not influence the push-out forces, connections where only the taper was sandblasted show higher push-out forces than taper connections without sandblasting or with only trunnion blasting. As the sandblasted tapers exhibit increased roughness parameters, a slight influence of the roughness of sandblasted tapers can be assumed. Nevertheless, the greatest bonding strength is achieved when both tapers are sandblasted, which is why further investigations should be conducted in this manner. In further investigations, adhesive bonded taper connections should be tested under more realistic conditions. Micromotion is the most common cause of failure of modular implant connections. These possible micromotions were not investigated in our study. This is the subject of further planned fatigue tests with 10 million load cycles under standardised testing conditions.

In the current study, we investigated taper combinations consisting of both partners being either TiAl6V6 ELI or CoCr28Mo6. Future investigations should also test the combination of these two materials. Furthermore, further research will investigate possible methods of revising of glued taper junctions.

## Conclusion

The push-out forces generated from the sandblasted tapers resulted in the highest push-out forces and were greater than push-out forces from mechanical assembled taper pairs reported in the literature. Therefore, the conclusion can be drawn that sandblasted and adhesive bonded taper connections are promising alternative for modular hip implant junctions. Slight differences in taper geometry are compensated by the adhesive and do not influence the push-out forces. Further investigation under realistic testing conditions is necessary to assess the suitability in the long-term.

## Supporting information

**S1 Table. Taper mismatch with corresponding push-out forces.** Overview over all 28 taper pairs with their push-out force and taper mismatch (angular difference). The colors indicate which taper was sandblasted: grey: outer taper sandblasted, light blue: trunnion sandblasted, dark blue: both sandblasted.
(DOCX)

**S1 Data.**
(XLSX)

**S2 Data.**
(XLSX)

## Author Contributions

**Conceptualization:** Beate Legutko, Philipp-Cornelius Pott.

**Data curation:** Ann-Kathrin Einfeldt, Beate Legutko.

**Formal analysis:** Ann-Kathrin Einfeldt, Beate Legutko.

**Funding acquisition:** Berend Denkena, Christof Hurschler, Bastian Welke.

**Investigation:** Ann-Kathrin Einfeldt, Beate Legutko, Philipp-Cornelius Pott.

**Supervision:** Bastian Welke.

**Writing – original draft:** Ann-Kathrin Einfeldt.

**Writing – review & editing:** Beate Legutko, Philipp-Cornelius Pott, Benjamin Bergmann, Berend Denkena, Christof Hurschler, Bastian Welke.

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
