## [Decision Letter · Decision Letter 0]

24 Apr 2024

PONE-D-24-03532Influence of the use of an adhesive connection on the joint strength of modular hip endoprosthesesPLOS ONE

Dear Dr. Welke,

Thank you for submitting your manuscript to PLOS ONE. After careful consideration, we feel that it has merit but does not fully meet PLOS ONE’s publication criteria as it currently stands. Therefore, we invite you to submit a revised version of the manuscript that addresses the points raised during the review process. **Please, address all the comments made by the reviewers.**

We look forward to receiving your revised manuscript.

Kind regards,

Antonio Riveiro Rodríguez, PhD

Academic Editor

PLOS ONE

Deutsche Forschungsgemeinschaft (DFG, German Research Foundation) within the SFB/TRR-298-SIIRI-Project-ID 426335750 “Safety integrated and infection reactive implants”.

Publication costs are covered by the German Research Foundation (DFG) and the Open Access Publication Fund of Hannover Medical School (MHH).

The authors would like to thank the Deutsche Forschungsgemeinschaft (DFG, German Research Foundation) for their financial support within the SFB/TRR-298-SIIRI-Project-ID 426335750 “Safety integrated and infection reactive implants” in subproject A04. 

Deutsche Forschungsgemeinschaft (DFG, German Research Foundation) within the SFB/TRR-298-SIIRI-Project-ID 426335750 “Safety integrated and infection reactive implants”.

Publication costs are covered by the German Research Foundation (DFG) and the Open Access Publication Fund of Hannover Medical School (MHH).

Reviewers' comments:

Reviewer's Responses to Questions

**Comments to the Author**

1. Is the manuscript technically sound, and do the data support the conclusions?

Reviewer #1: Yes

Reviewer #2: Partly

2. Has the statistical analysis been performed appropriately and rigorously? 

Reviewer #1: No

Reviewer #2: Yes

3. Have the authors made all data underlying the findings in their manuscript fully available?

Reviewer #1: Yes

Reviewer #2: Yes

4. Is the manuscript presented in an intelligible fashion and written in standard English?

Reviewer #1: Yes

Reviewer #2: Yes

5. Review Comments to the Author

**Reviewer #1:** The paper presents an interesting approach to secure taper connections in total hip arthroplasty against fretting and fretting-corrosion, respectively, via adhesive bonding of the connection. To do so, taper connections were fabricated, adhesively bonded and, subsequently, the resulting bonding strength was assessed in push-out tests. For inner tapers, a variety of different fabrication parameters was used which resulted in different surface topographies. The different topographies, however, did not influence the resulting bonding strength. In addition, one part or both parts of the taper connections were sandblasted, which influenced the resulting push-out forces.

It was, however, not clearly addressed, which taper connection would benefit from adhesive bonding, since modularity (aside from the better adjustment options) also allows the replacement of individual parts of the endoprosthesis in case of a revision. This would in my opinion be compromised by adhesive bonding unless the bonding could be unglued in case of a revision (?). This should definitely be addressed in the discussion of the study. Furthermore, it remained unclear to me, why so many different combinations processing parameters were used (Table 1 is rather confusing), especially since they did not really influence the resulting bonding strengths. Here, the paper would benefit from reducing the presented magnitude of different taper variations. Maybe the focus could be put on sandblasting vs. turned surfaces and the used materials?

Major revisions are, therefore, necessary before publication of the study.

Some more specific comments are the following:

Introduction:

As already pointed out, it remains unclear, which taper connection should or could be addressed with adhesive bonding approach. In the introduction you refer to studies regarding dual-neck total hip endoprosthesis, but these have widely disappeared from the market, as far as I know. The taper junction you actually tested appears to rather mimic the taper connection between head and stem taper (even though you used a twice as large taper angle).

Material and Methods:

Line 108: The taper angle is given as 5,665° but from Figure 1 it appears that this value refers to the half of the taper angle, which would result in a taper angle > 11°. Please be more accurate here and throughout the manuscript.

Line 114 -125: It is very hard for the reader to understand the combination of the process parameters used for manufacturing the specimens. In line 115 for example you write that two cutting speeds and three feeds were used, which would make six. From the table it becomes clear that not all parameters were combined with each other, but this does not become clear from the text. Also, it does not become clear, which material was used for which parameter combination and whether you produced inner and outer tapers from the two materials. Furthermore, it remains unclear how the different materials were combined for forming the taper connections. Did you just couple Ti-Ti and CoCr-CoCr? If so, why not couple CoCr-Ti as is done in many head-neck taper connections?

Line 180: How was the axial displacement determined (e.g. traverse position, optical measurement…)? I ask because I wonder whether the different inclinations are really caused by the “stiffness” of the connection as you write in line 318ff or if they reflect rather the deformation of the testing setup with increasing load levels.

Results

Table 2: Does the table refer to inner or outer tapers? How are the roughness values for the not sandblasted surfaces?

Figure 3: Does it refer to inner or outer tapers? Please add the information in the figure caption.

The term “Taper mismatch” is used in two different contents, in line 185 ff it refers to the deviation from the targeted value, later in the paper it refers to the difference between inner and outer taper. This is somehow confusing since “taper mismatch” is more commonly used synonymously to taper clearance (i.e. with respect to the difference between inner and outer taper).

Line 201-204: The slight differences in taper end diameters do not necessarily result in a changed taper contact situation but different taper angles do (see f.ex. Fig. 8 in Mueller et al., J Arthroplasty 2017, 32(10), 3191). Please use the difference between taper angles in order to characterize the contact (distal or proximal), since using end diameters is very confusing und unusual.

Line 245: Is the correlation statistically significant?

Line 268: If the adhesive is not observed after disconnecting the tapers, where do you think it is gone to? Is it likely that chips of dried adhesive have detached from the surface? If so, this would be a drawback since chips of the glue might act as three-body in the joint articulation or cause tissue reactions. This could be addressed in the discussion.

Discussion

Line 282f: See comment above (line 201-204). Please refer to the taper angle mismatch if discussing the contact situation.

**Reviewer #2: **This manuscript investigates the use of an adhesive and sandblasting on the potential stability of total hip head-neck modular junctions. The study is well-designed. The main criticisms focus on details in the manuscript that overstate the significance of how this study relates to hip modular junctions since the current application is still far removed, and the discussion of previous literature. Most suggestions are minor. In addition, please edit throughout for written English.

Abstract

The introduction focuses on micromotion as a failure mode. There is a disconnect between these statements and the paper's methods where push-out tests are used. No explanation is given for the disconnect. Please edit accordingly.

Introduction

Line 56: citation missing

Lines 65-66: Reference [3] is an investigation of bi-modular implants which have more than 2 parts, and is less relevant to the current study. Include more references here to make this point, there are plenty of investigations in the literature. Similarly, for the next sentence, the titanium neck adapter was actually TMZF, a different Ti-alloy than what is usually used, so the sentences are misleading.

Lines 81-82: This is not a relevant finding for the introduction. Taper connections of hip implants on the market have an average angular mismatch much smaller, on the order of 1 minute (.0167 degrees), so 0.096 would already be an outlier and is too large to be clinically relevant.

Suggest an overall re-writing of the literature background about factors that lead to damage in modular THAs. Only a few studies are cited, and they investigate a wide variety of implant types and dimensions that may or may not be relevant to the current study. The paragraphs do not summarize the literature from clinically relevant designs (the bimodular design referenced has been recalled, many studies of angular mismatch reference manufacturing tolerances that, although written in a standard, are much larger than what is actually used by manufacturers to create implants in clinical practice).

Materials and Methods

Table 1: The labeling of the samples is confusing. Makes it appears that all inner and outer tapers had the same material (Ti-alloy or CoCrMo alloy)? Why would a Ti-alloy taper pair be used when this is not relevant to total hip replacements? Suggest using more descriptive and intuitive labels.

Results

Lines 192 and 199: I don't know what mismatch means in this context since it is listed for the inner tapers or outer tapers. Difference from the intended value of 5.665? Mismatch usually refers to the difference between inner and outer tapers.

The combinations of pairs is very confusing, similar to my comment in the methods. Suggest adding a table that explicitly states which inner taper is paired with which outer taper and the resulting angular mismatch.

Figure 3: Please use a more descriptive caption. I assume these are all inner tapers because of the process parameters listed. Even with turning, these would still be considered very smooth inner tapers. Most stem tapers that are turned have trough to peak heights from 10 to 13 microns instead of 3 microns. This is fine, but worthy of a discussion point.

Line 242: Does the free volume take into account all surface differences (angular mismatch, sandblasting, turn machining parameter differences that results in greatly different surface contacts)? It does not seem surprising that there is no correlation between free volume and push-out.

Lines 256-261: not sure what the purpose of this paragraph is, steeper/not steeper is relevant to a stronger taper pair?

Discussion

Lines 272-274: what data is this discussion point referring to? It needs to be in the introduction if important.

Lines 278-281: Need to discuss the angular mismatch in relation to what is normal for actual THA. Many of the ones in this study would be outliers and not used in clinical practice therefore the conclusions drawn are not relevant/helpful

Lines 306-308: What does suitable mean in this context? What about the other studies in the literature? There have been EXTENSIVE studies with pull out/push out etc. The conversation about "suitable" is irrelevant to the paper. Recommend only saying that sandblasted surfaces resulted in the highest push-out forces that were greater than 2 kN.

The limitations section is not extensive enough. (lines 329-330). What are the more realistic conditions? What about realistic taper mismatch, etc?

Conclusion

Line 334-335: Please remove "generally accepted". There is no generally accepted value in this field.

All figures and tables: use better abbreviations or spell out entire words. Figures and tables are difficult to understand.

6. PLOS authors have the option to publish the peer review history of their article (what does this mean?). If published, this will include your full peer review and any attached files.

Reviewer #1: No

Reviewer #2: No

---

## [Author Response · Author response to Decision Letter 0]

22 Aug 2024

Dear Dr. Antonio Riveiro Rodríguez,

Thanks a lot for the opportunity to revise our paper in order to resubmit it to PLOS ONE. We have worked on all comments and have implemented all of the suggestions and points of criticism. The present response is done in a stepwise way regarding the reviewer’s comments (italic). Corresponding changes and additions have been highlighted in yellow in the “manuscript with track changes”.

Reviewer #1: The paper presents an interesting approach to secure taper connections in total hip arthroplasty against fretting and fretting-corrosion, respectively, via adhesive bonding of the connection. To do so, taper connections were fabricated, adhesively bonded and, subsequently, the resulting bonding strength was assessed in push-out tests. For inner tapers, a variety of different fabrication parameters was used which resulted in different surface topographies. The different topographies, however, did not influence the resulting bonding strength. In addition, one part or both parts of the taper connections were sandblasted, which influenced the resulting push-out forces.

It was, however, not clearly addressed, which taper connection would benefit from adhesive bonding, since modularity (aside from the better adjustment options) also allows the replacement of individual parts of the endoprosthesis in case of a revision. This would in my opinion be compromised by adhesive bonding unless the bonding could be unglued in case of a revision (?). This should definitely be addressed in the discussion of the study. 

Thank you very much for this important note. Our paper in fact focusses on the connection between the shank and a modular taper. Your comment regarding the replacement of the individual parts is really valuable and we added a section in the discussion (L347-350 in the revised manuscript). 

Furthermore, it remained unclear to me, why so many different combinations processing parameters were used (Table 1 is rather confusing), especially since they did not really influence the resulting bonding strengths. Here, the paper would benefit from reducing the presented magnitude of different taper variations. Maybe the focus could be put on sandblasting vs. turned surfaces and the used materials?

Thank you very much for this valuable comment. We agree that the table was quite confusing. We have simplified the table and described the sample groups in the text more clearly (L130-140). We nonetheless think that investigating different process parameters provides valuable information and that interestingly the different process parameters did not have an influence on the bonding strength which we did not anticipate.

Major revisions are, therefore, necessary before publication of the study.

Some more specific comments are the following:

Introduction:

As already pointed out, it remains unclear, which taper connection should or could be addressed with adhesive bonding approach. In the introduction you refer to studies regarding dual-neck total hip endoprosthesis, but these have widely disappeared from the market, as far as I know. The taper junction you actually tested appears to rather mimic the taper connection between head and stem taper (even though you used a twice as large taper angle).

Thank you very much for this note. As stated above, the paper focusses on the connection between the shank and a modular taper. We are aware of the disappearance of dual-taper junctions from the market because of the high revision rates. The motivation of this study was to investigate a possible improvement of the taper junctions which may allow dual-taper junctions to be safely used again, and in particular to exploit the flexibility in aligning the joint that they can provide in achieving an anatomically aligned joint reconstruction. We have supplemented some parts of the introduction to make this clearer to the reader.

Material and Methods:

Line 108: The taper angle is given as 5,665° but from Figure 1 it appears that this value refers to the half of the taper angle, which would result in a taper angle > 11°. Please be more accurate here and throughout the manuscript.

Thank you very much for this valuable comment. We have adjusted this accordingly so that the taper angle now has the correct value in the whole manuscript (L115-116).

The reviewer is absolutely right that the taper angle is >11°. We chose this larger angle to compare the results from this study with our preliminary study (18). In the preliminary study, we investigated the influence of the surface topography on the push-out force. The larger angle represents an unfavorable situation for the connection, both for the mechanical and for the bonded connection. We intentionally chose this in our project. Due to the larger taper angle, the average push-out force was only 1.8kN. In the current study, the same bonded taper junction achieved forces three times as high. In our view, this shows the potential of the bond.

Nevertheless, we are currently conducting a study in which we are investigating the cone connection with the taper angle of 12/14.

Line 114 -125: It is very hard for the reader to understand the combination of the process parameters used for manufacturing the specimens. In line 115 for example you write that two cutting speeds and three feeds were used, which would make six. From the table it becomes clear that not all parameters were combined with each other, but this does not become clear from the text. Also, it does not become clear, which material was used for which parameter combination and whether you produced inner and outer tapers from the two materials. Furthermore, it remains unclear how the different materials were combined for forming the taper connections. Did you just couple Ti-Ti and CoCr-CoCr? If so, why not couple CoCr-Ti as is done in many head-neck taper connections?

Thank you very much for this valuable comment. As we mentioned a previous comment, we have simplified the table and described the groups in the text more precisely (L130-140). Due to delivery bottlenecks in CoCr we did not have enough material to also test CoCr-Ti, and we will investigate this material pairing in further investigations. We have added a sentence in this regard to the discussion (L348-350).

Line 180: How was the axial displacement determined (e.g. traverse position, optical measurement…)? I ask because I wonder whether the different inclinations are really caused by the “stiffness” of the connection as you write in line 318ff or if they reflect rather the deformation of the testing setup with increasing load levels.

Thank you for this important observation. We in fact only recorded the displacement of the actuator of the materials testing machine. The recorded displacement was thus not corrected in the first version of the manuscript. In order to determine the influence of the setup on the measured displacement, we once again carried out tests using only the setup without samples. The upper stamp pressed directly on the lower structure. We determined a stiffness of 39.111 N/mm for the linear range. We then used this value to correct the measured displacement. In the manuscript, we replaced the diagram accordingly and added the following part (L274-275):

“The measured displacement of the testing machine was subsequently corrected for the stiffness of the setup.”

We have also added this description to the caption of Figure 6. 

Since it might be interesting for the reviewer, we show here the figure of how the graphs have changed after the stiffness correction

Results

Table 2: Does the table refer to inner or outer tapers? 

Thank you very much for this important observation. It was the trunnion, and we have added this to the table heading.

How are the roughness values for the not sandblasted surfaces?

Thank you very much for this comment. We added the values for the not sandblasted surfaces into the table.

Figure 3: Does it refer to inner or outer tapers? Please add the information in the figure caption.

Thank you very much again. Here as above it was the trunnion. We have added this to the figure heading.

The term “Taper mismatch” is used in two different contents, in line 185 ff it refers to the deviation from the targeted value, later in the paper it refers to the difference between inner and outer taper. This is somehow confusing since “taper mismatch” is more commonly used synonymously to taper clearance (i.e. with respect to the difference between inner and outer taper).

Thank you very much for this important comment. We have adapted the terms throughout the manuscript. We now use ‘nominal value’ instead of ‘target’. And we now only use mismatch to refer to the difference between trunnion and taper.

Line 201-204: The slight differences in taper end diameters do not necessarily result in a changed taper contact situation but different taper angles do (see f.ex. Fig. 8 in Mueller et al., J Arthroplasty 2017, 32(10), 3191). Please use the difference between taper angles in order to characterize the contact (distal or proximal), since using end diameters is very confusing und unusual.

We agree with the reviewer in this regard, we have referred to taper-mismatch to describe the difference in angle between the taper and trunnion and removed the first sentence referring to the end-diameters.

We added the following sentence to the manuscript (L211-214):

“All 28 taper pairs exhibit a negative taper mismatch (trunnion angle < taper angle) and therefore a proximal contact. All taper pairs show a taper mismatch greater than |0.1°| with a mean of -0.182 ± 0.047° (range -0.012° to -0.369°) (table in supplemental material).”

Line 245: Is the correlation statistically significant?

We have added the p-value for the correlations that are significant to the text (L259).

Line 268: If the adhesive is not observed after disconnecting the tapers, where do you think it is gone to? Is it likely that chips of dried adhesive have detached from the surface? If so, this would be a drawback since chips of the glue might act as three-body in the joint articulation or cause tissue reactions. This could be addressed in the discussion.

Thank you very much for this comment. The visual analysis showed that the adhesive adheres only to the sandblasted surfaces and not to the turned ones. So, in the case of sandblasting no chips of glue detached from the sandblasted surface. It will be important to investigate the system under dynamic loading to investigate if particles of dried adhesive are generated and escape into the joint.

Discussion

Line 282f: See comment above (line 201-204). Please refer to the taper angle mismatch if discussing the contact situation.

We agree with the reviewer as above, and therefore we changed the wording and removed the first sentence referring to the end-diameters (L282 first submission of the manuscript).

Reviewer #2: This manuscript investigates the use of an adhesive and sandblasting on the potential stability of total hip head-neck modular junctions. The study is well-designed. The main criticisms focus on details in the manuscript that overstate the significance of how this study relates to hip modular junctions since the current application is still far removed, and the discussion of previous literature. Most suggestions are minor. In addition, please edit throughout for written English.

Abstract

The introduction focuses on micromotion as a failure mode. There is a disconnect between these statements and the paper's methods where push-out tests are used. No explanation is given for the disconnect. Please edit accordingly.

Thank you for your comment, in response to this and the authors further comments regarding the introduction, we have significantly edited and reformulated the introduction to address this point. 

Introduction

Line 56: citation missing

Thank you very much for the noticing this, we have added an appropriate citation (L55).

Lines 65-66: Reference [3] is an investigation of bi-modular implants which have more than 2 parts, and is less relevant to the current study. Include more references here to make this point, there are plenty of investigations in the literature. Similarly, for the next sentence, the titanium neck adapter was actually TMZF, a different Ti-alloy than what is usually used, so the sentences are misleading.

The literature cited was chosen to generally illustrate the failure of a very promising technology, for hip replacement, bi-modular implants, and our relatively simple study investigating the effect of adhesive bonding on taper junction push-out force. We have attempted to address the reviewers well taken comments on the relevance of the cited literature and our study in significant revisions we have made to the introduction (as suggested by the reviewer below). 

Lines 81-82: This is not a relevant finding for the introduction. Taper connections of hip implants on the market have an average angular mismatch much smaller, on the order of 1 minute (.0167 degrees), so 0.096 would already be an outlier and is too large to be clinically relevant.

Thank you for this important comment, which we will attempt to address. It is true that taper mismatch of common stem and head tapers has recently been reported to range from about 0,025’ to -0,14’ (Mueller et al., 2017), which indeed is significantly less than the 0.096° (5.76’) we chose. Nonetheless, positive values that are significantly higher (for example 0.076° to 0.130° or 4.6’ to 7.8’ have also been reported), admittedly for an elongated taper design that is no longer available (Haschke et al., 2016). But more to the point, we have however oriented our study on the possible mismatch constellations that could occur in clinical settings as described by Ashkanfar et al. 2017 and Gustafson et al., 2023, who both conservatively estimated realistic possible taper mismatches based on the literature and their own retrievals to be as high as ±12’. The value we chose, about 6’ falls well within this range.

Suggest an overall re-writing of the literature background about factors that lead to damage in modular THAs. Only a few studies are cited, and they investigate a wide variety of implant types and dimensions that may or may not be relevant to the current study. The paragraphs do not summarize the literature from clinically relevant designs (the bimodular design referenced has been recalled, many studies of angular mismatch reference manufacturing tolerances that, although written in a standard, are much larger than what is actually used by manufacturers to create implants in clinical practice).

Thank you for your suggestion, we have significantly reformulated the introduction to better describe our motivation in relation to the literature. The intention of the introduction was not to provide a comprehensive overview of implant types and dimensions in general. Our motivation is based on the failure of bi-modular implants and their subsequent removal for the market, and our exploratory study of the effect of adhesive bonding on the most rudimentary test of junction stability; the push out test, which to our knowledge has not been investigated to date.

Materials and Methods

Table 1: The labeling of the samples is confusing. Makes it appears that all inner and outer tapers had the same material (Ti-alloy or CoCrMo alloy)? Why would a Ti-alloy taper pair be used when this is not relevant to total hip replacements? Suggest using more descriptive and intuitive labels.

Thank you very much for this important note. We see that the table was quite confusing and therefore we simplified the table and described the groups in the text more precisely. 

Results

Lines 192 and 199: I don't know what mismatch means in this context since it is listed for the inner tapers or outer tapers. Difference from the intended value of 5.665? Mismatch usually refers to the difference between inner and outer tapers.

Thank you very much for this important comment. We changed the wording in the section to taper “deviation from nominal value” instead of mismatch (L206-214). And we have added an additional table with the mismatch values to the supplement

The combinations of pairs is very confusing, similar to my comment in the methods. Suggest add

---

## [Decision Letter · Decision Letter 1]

9 Oct 2024

PONE-D-24-03532R1Influence of the use of an adhesive connection on the joint strength of modular hip endoprosthesesPLOS ONE

Dear Dr. Welke,

Thank you for submitting your manuscript to PLOS ONE. After careful consideration, we feel that it has merit but does not fully meet PLOS ONE’s publication criteria as it currently stands. Therefore, we invite you to submit a revised version of the manuscript that addresses the points raised during the review process. Please, address the minor comment made by reviewer 1 regarding Figure 4.

We look forward to receiving your revised manuscript.

Kind regards,

Antonio Riveiro Rodríguez, PhD

Academic Editor

PLOS ONE

Journal Requirements:

Reviewers' comments:

Reviewer's Responses to Questions

**Comments to the Author**

1. If the authors have adequately addressed your comments raised in a previous round of review and you feel that this manuscript is now acceptable for publication, you may indicate that here to bypass the “Comments to the Author” section, enter your conflict of interest statement in the “Confidential to Editor” section, and submit your "Accept" recommendation.

Reviewer #1: All comments have been addressed

Reviewer #3: (No Response)

2. Is the manuscript technically sound, and do the data support the conclusions?

Reviewer #1: Yes

Reviewer #3: Yes

3. Has the statistical analysis been performed appropriately and rigorously? 

Reviewer #1: Yes

Reviewer #3: Yes

4. Have the authors made all data underlying the findings in their manuscript fully available?

Reviewer #1: Yes

Reviewer #3: Yes

5. Is the manuscript presented in an intelligible fashion and written in standard English?

Reviewer #1: Yes

Reviewer #3: Yes

6. Review Comments to the Author

Reviewer #1: The authors have answered all comments satisfactorily. I have one more remark regarding Figure 4, please use the taper terms used in the text also for the figure (taper and trunnion instead of inner taper and outer taper). The manuscript can be accepted for publication now.

Reviewer #3: Your research and article are wonderful.

Please check the comments in the text of article and references.

7. PLOS authors have the option to publish the peer review history of their article (what does this mean?). If published, this will include your full peer review and any attached files.

Reviewer #1: No

Reviewer #3: **Yes: **Batoul Bagheripour

---

## [Author Response · Author response to Decision Letter 1]

11 Oct 2024

Dear Dr. Antonio Riveiro Rodríguez,

Thanks once again for the opportunity to revise our paper in order to resubmit it to PLOS ONE. 

We thank the reviewers for their work and have revised the manuscript and Figure 4 in accordance with the comments.

Reviewer #1: The authors have answered all comments satisfactorily. I have one more remark regarding Figure 4, please use the taper terms used in the text also for the figure (taper and trunnion instead of inner taper and outer taper). The manuscript can be accepted for publication now.

Many thanks for the comment. We have changed the caption for Figure 4 accordingly.

Reviewer #3: Please recheck the style of references. Did you use the Numbered or Vancouver?

Thank you for pointing this out, we apologize for this inadvertence. We have adjusted the references and citations in accordance with the journal's specifications.

Reviewer #3: Please write your hypothesis here.

On recommendation, we have added the following sentence at the end of the introduction (L108-109):

“We hypothesize that a bonded connection with a rough surface achieves at least the same strength as conventionally joined implant junctions.”

Reviewer #3: Please check references 11 and 13. No pages are listed for these articles.

Many thanks for the hint. We have added the information for the page numbers

---

## [Decision Letter · Decision Letter 2]

4 Nov 2024

Influence of the use of an adhesive connection on the joint strength of modular hip endoprostheses

PONE-D-24-03532R2

Dear Dr. Welke,

We’re pleased to inform you that your manuscript has been judged scientifically suitable for publication and will be formally accepted for publication once it meets all outstanding technical requirements.

Kind regards,

Antonio Riveiro Rodríguez, PhD

Academic Editor

PLOS ONE

Additional Editor Comments (optional):

Reviewers' comments:

Reviewer's Responses to Questions

**Comments to the Author**

1. If the authors have adequately addressed your comments raised in a previous round of review and you feel that this manuscript is now acceptable for publication, you may indicate that here to bypass the “Comments to the Author” section, enter your conflict of interest statement in the “Confidential to Editor” section, and submit your "Accept" recommendation.

Reviewer #1: All comments have been addressed

2. Is the manuscript technically sound, and do the data support the conclusions?

Reviewer #1: Yes

3. Has the statistical analysis been performed appropriately and rigorously? 

Reviewer #1: Yes

4. Have the authors made all data underlying the findings in their manuscript fully available?

Reviewer #1: Yes

5. Is the manuscript presented in an intelligible fashion and written in standard English?

Reviewer #1: Yes

6. Review Comments to the Author

Reviewer #1: (No Response)

7. PLOS authors have the option to publish the peer review history of their article (what does this mean?). If published, this will include your full peer review and any attached files.

Reviewer #1: No

---

## [Editor Report · Acceptance letter]

7 Nov 2024

PONE-D-24-03532R2 

PLOS ONE

Dear Dr. Welke, 

I'm pleased to inform you that your manuscript has been deemed suitable for publication in PLOS ONE. Congratulations! Your manuscript is now being handed over to our production team.

Kind regards, 

on behalf of

Dr. Antonio Riveiro Rodríguez 

Academic Editor

PLOS ONE